# Research on Measurement Method of Parachute Scanning Platform Based on MEMS Device

**DOI:** 10.3390/mi12040402

**Published:** 2021-04-05

**Authors:** Ning Liu, Tianqi Tian, Zhong Su, Wenhao Qi

**Affiliations:** 1Beijing Key Laboratory of High Dynamic Navigation Technology, Beijing Information Science and Technological University, Beijing 100101, China; tianqitian@foxmail.com (T.T.); sz@bistu.edu.cn (Z.S.); qwh18769411875@foxmail.com (W.Q.); 2School of Automation, Beijing Institute of Technology, Beijing 100081, China

**Keywords:** MEMS Gyro, attitude measurement, extended Kalman filter, terminal-sensitive projectile, parachute scanning platform

## Abstract

This paper studies the measurement of motion parameters of a parachute scanning platform. The movement of a parachute scanning platform has fast rotational velocity and a complex attitude. Therefore, traditional measurement methods cannot measure the motion parameters accurately, and thus fail to satisfy the requirements for the measurement of parachute scanning platform motion parameters. In order to solve these problems, a method for measuring the motion parameters of a parachute scanning platform based on a combination of magnetic and inertial sensors is proposed in this paper. First, scanning motion characteristics of a parachute-terminal-sensitive projectile are analyzed. Next, a high-precision parachute scanning platform attitude measurement device is designed to obtain the data of magnetic and inertial sensors. Then the extended Kalman filter is used to filter and observe errors. The scanning angle, the scanning angle velocity, the falling velocity, and the 2D scanning attitude are obtained. Finally, the accuracy and feasibility of the algorithm are analyzed and validated by MATLAB simulation, semi-physical simulation, and airdrop experiments. The presented research results can provide helpful references for the design and analysis of parachute scanning platforms, which can reduce development time and cost.

## 1. Introduction

As an important air-to-ground detection platform, the parachute scanning platform has been widely applied to many fields, including air-to-ground weapons and aerial detection. Its movement has the characteristics of high speed and complex attitude. The movement state of the platform has an important impact on its detection quality and efficiency during the fall of the platform. Many studies assume a pendulum motion rather than a scanning motion [1,2]. However, a pendulum motion will make the parachute platform detect the same area repeatedly, which is not necessary in some situations. In order to expand the detection area of the parachute platform and avoid repeated detection, it is better to make the parachute platform spiral scanning downward. Therefore, research on the measurement methods of the motion parameters of a parachute scanning platform is of great significance for improving its detection performance. This paper uses a terminal sensitive projectile that is a typical air-to-ground scanning weapon as a research object.

With the continuous development of modern armor defense technology, many types of defensive materials and technologies with excellent performance have been applied to tanks, which has greatly improved their defense capabilities [3]. In response to the high defensive performance of tanks, anti-tank weapons have also been developed rapidly. Long-distance, high-precision, low-cost, high-efficiency, and cost-effective terminal-sensitive projectiles are favored by many countries. The United States, Germany, France, Sweden, and Russia have all conducted research on terminal-sensitive projectiles, and in this field, they are in a leading position [4,5].

A terminal-sensitive projectile is a type of smart and flexible ammunition that can identify and strike the target automatically after being thrown and has high efficiency [6]. Generally, a terminal-sensitive projectile is carried to a designated area by a carrier, such as a missile, a rocket, a projectile, or a spreader, and a number of terminal-sensitive projectiles are thrown according to a planned pattern. After a terminal-sensitive projectile is thrown, it will go through three phases, namely, the free-fall phase, the deceleration and reducing rotation phase, and the steady-state scanning phase. During the steady-state scanning phase, the terminal-sensitive projectile scans the designated area, and the projectile quickly strikes the armored vehicle once it is detected [7]. The motion parameters and their changing laws in the steady-state scanning stage directly influence the hit probability and the damage probability of a terminal-sensitive projectile [8]. Therefore, it is of great significance to research the changing law of motion parameters of a parachute scanning platform. The main research objective regarding the terminal-sensitive projectile scanning platform’s motion is to achieve real-time measurement of the motion parameters, such as the steady-state scanning angle, the scanning angular velocity, the falling velocity, and the 2D scanning attitude [9].

Much research has been conducted on methods for the accurate determination of motion parameters of a terminal-sensitive projectile. Under the assumption that the surrounding geomagnetic field is constant, Zhou et al. [10] proposed a method for attitude measurement of a terminal-sensitive projectile based on magnetic sensor data. However, in practice, the surrounding geomagnetic field can be easily changed by external disturbances. Therefore, the attitude measurement method based only on the geomagnetic field cannot guarantee high accuracy. Ma [11] from North University of China analyzed the measurement method of stable scanning parameters of a terminal-sensitive projectile based on miniature inertial measurement units (MIMUs). The analysis results showed that the accuracy of the formula approximation method was better than that of the integral solution algorithm. However, the inertial navigation measurement requires information on the initial attitude angle, and initial errors can easily appear. In addition, the errors in the solution process accumulate with the integral, so they increase with time. Cai [12] proposed a method based on 12 accelerometers and a magnetic sensor for the attitude measurement of rotating projectiles. However, since the attitude angles were obtained by the accelerometers and magnetic sensor, high solution accuracy was difficult to achieve. Li et al. [13] used contrastive simulation to validate the quaternion transformation model and its advantages in describing the revolving motion of the terminal sensing ammunition. The simulated results showed that the ballistic model amended by the quaternion method is more suitable for describing the terminal sensing ammunition’ s scanning motion with a large or uncertain angle than the Euler equation.

The traditional attitude measurement methods often use a single sensor to measure the target information but cannot adapt to various complex environments and can be easily interfered with [14,15]. The multi-sensor data fusion strategy can solve the problem of the low accuracy of a single-sensor measurement in extreme environments [16,17]. At the same time, the reliability, adaptability, and resolution of a multi-sensor measurement system are better than those of a single-sensor measurement system when the performance of the multi-sensor device is at the same level as that of the single sensor device [18,19]. Aiming at addressing the mentioned problems, this paper proposes a method of motion parameter measuring a parachute scanning platform by using three-axis magnetic sensors, a three-axis gyroscope, and a three-axis accelerometer. The proposed method can solve the problems of easy interference of the geomagnetic information and cumulative error of the inertial measurement. The proposed method has the advantages of wide adaptability, high measurement precision, and anti-magnetic interference.

The proposed attitude measurement method is evaluated by taking a terminal-sensitive projectile as an example. First, the scanning motion characteristics of a parachute terminal-sensitive projectile are analyzed. Next, the combination of magnetic sensors and inertial sensors is used to measure the motion parameters of the terminal-sensitive projectile. Then the extended Kalman filter (EKF) is used to observe and filter the measurement error, and the scanning angle, scanning angular velocity, falling velocity, and 2D scanning attitude are obtained. Finally, the semi-physical turntable simulation experiment and the airdrop experiment are carried out, and the experimental results verify the feasibility and accuracy of the proposed attitude measurement method. In order to obtain more accurate geomagnetic and inertial sensor data, a symmetrical parachute scanning platform attitude measuring device that can effectively reduce the measurement error is designed. The contribution of this work is twofold: A reliable attitude measurement method of parachute scanning platforms is proposed, and the presented results can provide helpful references for the design and analysis of parachute scanning platforms. Moreover, accurate scanning parameters of the test samples can help to improve the design efficiency and reduce the numbers of parameter adjustment tests, test samples, and labor and time costs, thereby saving both time and costs.

The rest of this paper is organized as follows. In Section 2, the principle of attitude measurement is explained. The design process of the attitude measurement algorithm and the data fusion based on the EKF are introduced in Section 3. In Section 4, the simulation on the geomagnetic field and the semi-physical turntable simulation are presented to verify the feasibility of the proposed method and the accuracy of the solution results. The algorithm is further verified by the airdrop experiment in Section 5. Finally, in Section 6, the main conclusions are drawn and future research directions are given.

## 2. Attitude Measurement

### 2.1. System Composition and Working Principle of Attitude Measurement Device

In order to obtain the flight attitude, rotational velocity, falling velocity, and 2D scanning attitude of a terminal-sensitive projectile, a parachute scanning platform attitude measuring device is designed. This device can not only record measurement data and process data of each sensor during the motion of a parachute terminal-sensitive projectile but can also significantly reduce the measurement error and compensate for errors of collected data due to the position layout design and post-processing of each sensor module on the circuit board. In addition, the real data used in this study are collected by this device.

#### 2.1.1. Designed System Composition and Working Principle

The attitude measuring device is presented in Figure 1, where it can be seen that it mainly consists of the geomagnetic data acquisition module, gyroscope data acquisition module, acceleration data acquisition module, data processing module, data recording module, power supply module, and communication module.

The geomagnetic data acquisition module is responsible for the collection of geomagnetic data in three directions. The gyroscope data acquisition module is responsible for collecting angle velocity data during the flight. Furthermore, the acceleration data acquisition module is responsible for collecting accelerometer data during the flight. The data processing module processes the data collected from the three mentioned sensors and outputs the information on the terminal-sensitive projectile, such as flight attitude and falling velocity. The data recording module receives the information from the data processing module and stores it at high speed. The communication module is not only used to read the data from the data recording module and upload it to a computer but can also receive control instructions from the computer and transmit them to the data processing module. Lastly, the power module is responsible for providing power to the device.

#### 2.1.2. Attitude Measuring Device Characteristics

In order to meet the performance requirements of a parachute scanning platform attitude measurement device, the optimal selection and layout of each sensor are determined. The system error is eliminated by optimizing the sensor layout. The magnetic sensor with high anti-overload and sensitivity and large-range characteristics is orthogonally placed and fixed to the projectile. According to the angular velocity range, a gyroscope with a suitable measurement range, having high anti-overload, low cross-coupling, and wide temperature properties, are selected and it is orthogonally placed and fixed to the projectile. The selection and layout of each sensor are described in detail in Section 5.

### 2.2. Attitude Algorithm Principle

#### 2.2.1. Coordinate System and Scanning Motion

The terminal-sensitive projectile is ammunition that can automatically find a target by motion scanning. As mentioned previously, its movement process can be divided into three phases: the free-fall phase, deceleration and reducing rotation phase, and steady-state scanning phase. In addition, the hit probability and the damage probability mainly depend on the scanning angle, scanning angular velocity, and falling velocity in the steady-state scanning phase. It should be noted that this paper studies the motion parameter measurement of a terminal-sensitive projectile in the steady-state scanning phase, which can provide necessary data support for the analysis and design of the motion parameters of a terminal-sensitive projectile in the steady-state scanning phase.

The terminal-sensitive projectile is usually transported by the carrier above the target area for projection. In this paper, the coordinate system mainly includes two coordinate systems, which are the projectile coordinate system and the navigation coordinate system. The illustration of the coordinate system and terminal-sensitive projectile’s motion process is shown in Figure 2. The local geographic coordinate system (North–East–Down) is used as a navigation coordinate system, where the x axis points to the true north, the *y* axis points to the east, and the *z* axis points vertically to the ground. The projectile coordinate system is constructed such that the origin is as at the center of mass of the projectile. The $z_{b}$ axis points to the upside along the axial direction of the projectile, the $x_{b}$ axis is perpendicular to the $z_{b}$ axis along the radial direction of the projectile, and the $y_{b}$ axis is perpendicular to the $x_{b}o_{b}z_{b}$ plane along the radial direction, following the right-hand rule. In order to accurately describe the projectile attitude in the navigation coordinate system, it is assumed that the angle formed by rotating around the $x_{n}$ axis is the roll angle ϕ, the angle formed by rotating around the $y_{e}$ axis is the pitch angle θ, and the angle formed by rotating around the $z_{d}$ axis is the yaw angle ψ, as shown in Figure 2.

After the pitch angle and roll angle are determined, the scanning angle and scanning angular velocity of the terminal-sensitive projectile can be respectively obtained as follows:(1)$$\{\begin{array}{c} \alpha =\sqrt[{2}]{\phi^{2}+\theta^{2}}\\ \omega =\frac{G_{z}}{\cos\alpha \cdot 360} \end{array}$$
where α denotes the scanning angle, ω denotes the scanning angular velocity, and $G_{z}$ is the output value of the gyroscope in the $z_{b}$-axis direction.

#### 2.2.2. Quaternion Principle

The quaternion has the advantages of simple calculation and small storage space required, and the orthogonality of the quaternion attitude matrix can be ensured by simple orthogonal processing. Due to its characteristics, it has been widely applied to coordinate transformation and description of rigid body rotation motion. A quaternion represents a super complex number consisting of one real unit and three imaginary units. In order to apply the quaternion to the transformation from the navigation coordinate system to the projectile coordinate system, a unit quaternion is defined as follows: $Q=q_{0}+{q_{0}}^{i}{{+q}_{0}}^{j}{{+q}_{0}}^{k}$. Correspondingly, a rotation operator $Q\cdot Q^{T}$ is introduced, and it can perform rotation transformation on any vector without changing its size during coordinate transformation and attitude description [20].

## 3. Motion Parameter-Solving Algorithm

The motion parameters of a terminal-sensitive projectile in the steady-state scanning phase are measured by the three-axis gyroscope, three-axis accelerometer, and three-axis magnetic sensor. A flow diagram of the motion parameter-solving algorithm is shown in Figure 3. As shown in Figure 3, first the bomb-mounted computer obtains the output data of the three-axis gyroscope, three-axis accelerometer, and three-axis magnetic sensor. Next, the quaternions and EKF are used for attitude and data fusion. Then the attitude and velocity are obtained. Finally, the accurate scanning motion parameters, such as the scanning angle and scanning angular, are obtained by (1). In Figure 3, $\omega_{x}$, $\omega_{y}$ and $\omega_{z}$ denote the gyroscope output data in the x, y and *z* directions, respectively; $a_{x}$, $a_{y}$ and $a_{z}$ denote the accelerometer output data in the x, y and z directions, respectively; and $m_{x}$, $m_{y}$ and $m_{z}$ denote the output data of the magnetic sensor in the x, y and z directions, respectively.

### 3.1. Attitude Measuring by Inertial Navigation Device

#### 3.1.1. Initial Alignment

The initial attitude alignment is required before the attitude measuring. Since an unmanned aerial vehicle (UAV) is in a stable state of hovering before the projectile is thrown, it is a good practice to use the data within 20 ms before the device is thrown for initial alignment. The initial alignment formulas of roll angle ϕ and pitch angle θ can be respectively expressed as
(2)$$\phi_{0}=\arctan\frac{a_{x}{}_{0}}{a_{z}{}_{0}}$$
(3)$$\theta_{0}=\arcsin\frac{a_{y0}}{\sqrt[{2}]{a_{x0}{}^{2}+a_{y0}{}^{2}+a_{z0}{}^{2}}}$$
where $\phi_{0}$ denotes the initial roll angle, $\theta_{0}$ represents the initial pitch angle, and $a_{x0}$, $a_{y0}$ and $a_{z0}$ are the average values of the output data of the three-axis accelerometer within the first 20 ms (stationary state) in the x, y, and z directions, respectively.

When the roll angle and the pitch angle are known, the initial magnetic heading angle $\psi_{0}$ can be obtained as follows:(4)$$\psi_{0}=\arctan2(M_{y},M_{x})+D$$
(5)$$M_{x}=m_{x0}\cos\theta +m_{y0}\sin\theta \sin\phi +m_{z0}\cos\theta \sin\phi ,$$
(6)$$M_{y}=m_{y0}\cos\phi -m_{z0}\sin\phi$$
where D denotes the local magnetic declination and $m_{x0}$, $m_{y0}$ and $m_{z0}$ denote the average values of the output data of the three-axis magnetic sensor within 20 ms in the three directions before the device is thrown, respectively.

#### 3.1.2. Attitude Calculation

(i) The initial attitude angle, which is also called the Euler angle, is the basis of the initial attitude matrix. The direction cosine matrix $C_{b}^{n}$ from the carrier system *b* to the navigation coordinate system *n* can be expressed as
(7)$$\begin{array}{c} C_{b}^{n}=\left[\begin{array}{ccc} c_{11} & c_{12} & c_{13}\\ c_{21} & c_{22} & c_{23}\\ c_{31} & c_{32} & c_{33} \end{array}\right]=\\ \left[\begin{array}{ccc} \cos\theta \cos\psi & -\cos\phi \sin\psi +\sin\phi \sin\theta \cos\psi & \sin\phi \sin\psi +\cos\phi \sin\theta \cos\psi \\ \cos\theta \sin\psi & \cos\phi \cos\psi +\sin\phi \sin\theta \sin\psi & -\sin\phi \cos\psi +\cos\phi \sin\theta \sin\psi \\ -\sin\theta & \sin\phi \cos\theta & \cos\phi \cos\theta \end{array}\right]. \end{array}$$

(ii) After the initial direction cosine matrix $C_{b}^{n}$ is obtained by Equation (6), the initial quaternion can be obtained by:(8)$$q_{0}={\left[\begin{array}{cccc} a & b & c & d \end{array}\right]}^{T}$$
(9)$$a=\frac{1}{2}{\left(1{+c}_{11}{+c}_{22}{+c}_{33}\right)}^{1/2}$$
(10)$$b=\frac{1}{4a}(c_{32}-c_{23})$$
(11)$$c=\frac{1}{4a}(c_{13}-c_{31})$$
(12)$$d=\frac{1}{4a}(c_{21}-c_{12})$$

(iii) The quaternion is solved by the time-sampled incremental method of the Pica solution. The quaternion is updated by
(13)$$q_{k+1}=\left\{\cos\frac{\Delta \theta_{k+1}}{2}\cdot I+\frac{2\cdot \sin\frac{\Delta \theta_{k+1}}{2}}{\Delta \theta_{k+1}}\left[\Delta \vartheta \right]\right\}q_{k}$$
where I denotes a 4 × 4 unit matrix; $\Delta \theta_{k+1}=\sqrt{\omega_{x}^{2}+\omega_{y}^{2}+\omega_{z}^{2}}\cdot \Delta t$;
(14)$$\Delta \vartheta_{k+1}={\left[\begin{array}{cccc} 0 & \omega_{z} & -\omega_{y} & \omega_{x}\\ -\omega_{z} & 0 & \omega_{x} & \omega_{y}\\ \omega_{y} & -\omega_{x} & 0 & \omega_{z}\\ -\omega_{x} & -\omega_{y} & -\omega_{z} & 0 \end{array}\right]}_{k+1}\cdot \frac{\Delta t}{2};\;\Delta t=\;0.001\;s\;\mathrm{is}\;\mathrm{sampling}\;\mathrm{period}.$$

(iv) After the quaternion is updated by Equation (9), the Euler angle can be obtained by
(15)$$\phi_{k+1}={\left[\frac{2(cd+ab)}{\left(a^{2}-b^{2}-c^{2}+d^{2}\right)}\right]}_{k+1}$$
(16)$$\theta_{k+1}={\left[2\left(ac-bd\right)\right]}_{k+1}$$
(17)$$\psi_{k+1}={\left[\frac{2\left(bc+ad\right)}{a^{2}+b^{2}-c^{2}-d^{2}}\right]}_{k+1}$$

Following the above steps, the initial attitude alignment, attitude matrix update, quaternion update, and attitude update are performed. In this way, the attitude data of the entire steady-state scanning phase can be obtained [21,22].

#### 3.1.3. Position and Falling Velocity Calculation

The initial values of velocity, position, and acceleration are assumed to be set as follows: $v_{0}=[\begin{array}{ccc} 0 & 0 & 0 \end{array}]$, $P_{0}=[\begin{array}{ccc} 0 & 0 & 0 \end{array}]$, and $a_{0}=[\begin{array}{ccc} 0 & 0 & g \end{array}]$, respectively. After the output data of the accelerometer are filtered by the EKF, the velocity and position of the terminal-sensitive projectile are respectively obtained by
(18)$$v_{k+1}=v_{k}+\int_{t_{k}}^{t_{k+}{}_{1}}\frac{1}{2}\cdot (a_{k}+a_{k+1})dt$$
(19)$$P_{k+1}=P_{k}+v_{k}\Delta t+\frac{1}{2^{2}}(a_{k}+a_{k+1})\Delta t^{2}$$
where $v_{k+1}$ represents the velocity at time (*k* + 1), *P_k_*_+1_ is the position at time (*k* + 1), Δt=0.001 s is the sampling period, and $a_{k+1}$ is the acceleration at time (*k* + 1).

### 3.2. Attitude Measurement Based on Geomagnetic Field

#### 3.2.1. Attitude Measurement Model Based on Geomagnetic Field

The terminal-sensitive projectile is usually dropped by a dispenser about 300 m above the ground, and after launching, it goes through the free-fall phase, deceleration and reducing rotation phase, and steady-state scanning phase. During the steady-state scanning phase, the terminal-sensitive projectile scans the designated area and quickly strikes the armored vehicle once the target has been detected. During the falling phase of the terminal-sensitive projectile, its spatial displacement is small, and the geomagnetic field hardly changes. Therefore, it can be considered that the geomagnetic field is constant during the falling phase of the terminal-sensitive projectile.

In order to facilitate model development, the falling velocity of the terminal-sensitive projectile is ignored. A simplified steady-state scanning model of the terminal-sensitive projectile is built according to the rotation cycle of the scanning motion of a terminal-sensitive projectile. The scanning motion diagram is presented in Figure 4.

The geomagnetic model of the terminal-sensitive projectile motion is built using the navigation coordinate system $o_{-}x_{n}y_{e}z_{d}$, as shown in Figure 4. In Figure 4, γ denotes the magnetic slant angle, B represents the geomagnetic field strength, and β denotes the phase angle.

The total strength of the geomagnetic field *B* represents the vector sum of the magnetic field strengths in the three directions. Under the premise of neglecting the change in the geomagnetic field, the total strength of the geomagnetic field B and the magnetic slant angle γ can be considered constants. In the navigation coordinate system, the strength of the geomagnetic field in the three directions can be obtained by
(20)$$\left\{ \begin{array}{l} Bx=-B\cdot \cos\gamma \\ By=0\\ Bz=B\cdot \sin\gamma \end{array}\right.$$

The geomagnetic field strength in the three directions of the projectile coordinate system can be obtained by
(21)$$\left\{ \begin{array}{l} m_{xb}=-B(\cos\alpha \cos(\omega t+\beta )\cos\gamma +\sin\alpha \sin\gamma )\\ m_{yb}=-B\cos\gamma \sin(\omega t+\beta )\\ m_{zb}=B(\sin\alpha \cos(\omega t+\beta )\cos\gamma -\sin\gamma \sin\alpha ) \end{array}\right.$$
where the geomagnetic field strength is *B* = 0.6 gauss, the data sampling period is Δt = 0.001 s, the scanning angle velocity is ω=4 rev/s, the initial phase angle is $\beta =0^{o}$, the scanning angle is $\alpha =30^{o}$, the magnetic slant angle is $\gamma =30^{o}$, *t* denotes time, and $m_{xb}$, $m_{yb}$ and $m_{zb}$ are the geomagnetic field strengths of the projectile coordinate system in the $x_{b}$, $y_{b}$ and $z_{b}$ directions, respectively.

#### 3.2.2. Attitude Calculation

The scanning angular velocity ω and the scanning angle α can be respectively obtained by
(22)$$\{\begin{array}{c} \omega =\frac{1}{t}(\arcsin(-\frac{m_{yb}}{\cos\gamma \sqrt{m_{xb}^{2}+m_{yb}^{2}+m_{zb}^{2}}})-\beta ){}^{}{}^{}\\ \alpha =\arcsin(\frac{B\cdot \sin\gamma }{\sqrt{m_{xb}^{2}+m_{zb}^{2}}})+\arccos(\frac{m_{xb}}{\sqrt{m_{xb}^{2}+m_{zb}^{2}}})) \end{array}$$
where the range of the scan angle α is [0, π/2].

### 3.3. Combined Filter

The data measured by the sensor contains certain noise. Filtering the measured data using a suitable filter helps to improve the accuracy of the attitude result. The EKF has the characteristics of the simple algorithm and calculation and is suitable for the Gaussian environment of a terminal-sensitive projectile.

In this paper, the specific filtering process of the combined filter is as follows. First, the position, velocity, attitude errors and output errors of the inertial device are used as state variables of the EKF. Next, a 15-dimensional filter model is established according to the motion characteristics of the terminal-sensitive projectile in the steady-state scanning phase. Then the position and attitude measurement errors are used as observation variables of the system. Finally, more accurate scanning parameters are obtained.

#### 3.3.1. State Variable Selection and State Equation Establishment

In this paper, the following errors are used as state variables of the EKF, which can be expressed as
(23)$$\begin{array}{l} X=[\Delta P_{x},\Delta P_{y},\Delta P_{z},\Delta v_{x},\Delta v_{y},\Delta v_{z},\Delta \phi ,\Delta \theta \\ {}_{}{}_{},\Delta \psi ,\Delta \alpha_{x},\Delta \alpha_{y},\Delta \alpha_{z},\Delta \omega_{x},\Delta \omega_{y},\Delta \omega_{z}] \end{array}$$
where $\Delta P_{x}$, $\Delta P_{y}$ and $\Delta P_{z}$ denote the position errors in the *x*, *y* and *z* directions, respectively; $\Delta v_{x}$, $\Delta v_{y}$ and $\Delta v_{z}$ denote the velocity errors in the *x*, *y* and *z* directions, respectively; Δϕ, Δθ and Δψ represent the roll angle error, pitch angle error, and heading angle error, respectively; $\Delta \alpha_{x}$, $\Delta \alpha_{y}$ and $\Delta \alpha_{z}$ are the accelerometer zero-offset errors in the *x*, *y* and *z* directions, respectively; and lastly, $\Delta \omega_{x}$, $\Delta \omega_{y}$, and $\Delta \omega_{z}$ are the gyroscope zero-offset errors in the *x*, *y* and *z* directions, respectively.

The discrete-time state equation of the system based on the state variables can be expressed as
(24)$$\{\begin{array}{l} \Delta P_{k+1}=\Delta P_{k}+\Delta v_{k}\cdot \Delta t\\ \Delta v_{k+1}=\Delta v_{k}+C_{b}^{n}\Delta a_{k}\cdot \Delta t+St\times \Delta \varphi_{k}\cdot \Delta t\\ \Delta \varphi_{k+1}=\Delta \varphi_{k}-C_{b}^{n}\Delta \omega_{k}\cdot \Delta t\\ \Delta a_{k+1}=\Delta a_{k}B_{1}+W_{k\cdot a}\\ \Delta \omega_{k+1}=\Delta \omega_{k}B_{2}+W_{k\cdot \omega } \end{array}$$
where St denotes the askew symmetric matrix $St=\left[\begin{array}{ccc} 0 & -a_{z} & a_{y}\\ a_{z} & 0 & -a_{x}\\ -a_{y} & a_{x} & 0 \end{array}\right]$; $\Delta P_{k+1}$ is the position error at time (*k* + 1); $\Delta v_{k+1}$ is the velocity error at time (*k* + 1); $\Delta \varphi_{k+1}$ is the attitude error at time (*k* + 1); $\Delta a_{k+1}$ is the accelerometer zero-offset error at time (*k* + 1); $\Delta \omega_{k+1}$ is the gyro zero-offset error at time (*k* + 1); $B_{1}$ and $B_{2}$ are the scaling factors of the accelerometer zero-offset error and the gyroscope zero-offset error, respectively, and according to the analysis of experimental data, they are very small; $W_{k\cdot a}$ and $W_{k\cdot \omega }$ are random system dynamic noises of accelerometer and gyroscope errors at time *k*, respectively; and the mean and variance of the noise satisfy the zero-mean white noise sequence of $E\left[W_{k}\right]=0,E\left[W_{k}W_{j}^{T}\right]{|}_{k=j}=Q_{k},E\left[W_{k}W_{j}^{T}\right]{|}_{k\neq j}=0$.

#### 3.3.2. Observation Variable Selection and Observation Equation Establishment

The position error and attitude measurement error are used as observation variables of the system. The attitude measurement error is obtained from the geomagnetic information and inertial information. The set of the observation equations can be expressed as follows:(25)$$Z=\left[\begin{array}{c} P_{x}-P_{IMUx}\\ \begin{array}{c} \begin{array}{c} P_{y}-P_{IMUy}\\ P_{z}-P_{IMUz}\\ \phi_{M}-\phi_{IMU} \end{array}\\ \theta_{M}-\theta_{IMU}\\ \psi_{M}-\psi_{IMU} \end{array} \end{array}\right]$$
where $P_{x},P_{y},P_{z}$ denote the position information obtained by the ideal motion state and coordinate values of the landing point; $P_{IMUx},P_{IMUy},P_{IMUz}$ denote the position information of the terminal-sensitive projectile obtained from the inertial device data; $\phi_{M},\theta_{M},\psi_{M}$ are the roll, pitch, and yaw angles obtained by the geomagnetic information, respectively; and lastly, $\phi_{IMU},\theta_{IMU},\psi_{IMU}$ are the roll, pitch, and yaw angles obtained by the inertial device data, respectively.

The observation equation of the EKF can be expressed as
(26)$$Z_{k}=h[X_{k}]+V_{k}$$
where $Z_{k}$ denoted the observed variable, $V_{k}$ is the noise, and the covariance matrix of the noise is expressed as $R_{k}=E[V_{k}V_{k}^{T}]$.

#### 3.3.3. Filtering Algorithm

The EKF is suitable for geomagnetic and inertial sensor integrated measurement. The state and observation equations of the system are given in Equations (17) and (18), respectively. The main process of the EKF is as follows.

Step 1: The initial state: attitude error, position error, and velocity error:(27)$$X_{0}=\left[\begin{array}{c} 0.01\cdot I_{3}\\ 0.0001\cdot I_{3}\\ 0.01\cdot \pi /180\cdot I_{3}\\ 0.3\cdot I_{3}\\ 0.5\cdot \pi /180\cdot I_{3} \end{array}\right]$$

Step 2: State prediction:(28)$$X_{k+1|k}=f\left[X_{k}\right]+W_{k+1}$$

Step 3: Observation prediction:(29)$$Z_{k+1|k}=h[X_{k+1|k}]+V_{k+1}$$

Step 4: Covariance matrix prediction
(30)$$P_{k+1|k}=\Phi_{k+1}P_{k|k}\Phi_{k+1}^{T}+Q_{k+1}$$

Step 5: Calculation of the Kalman gain
(31)$$K_{k+1}=P_{k+1|k}H_{k+1}^{T}(H_{k+1}P_{k+1|k}H_{k+1}^{T}+R_{k+1}){}^{-1}$$

Step 6: State variable update
(32)$$X_{k+1}=X_{k+1|k}+K_{k+1}(Z_{k+1}-Z_{k+1|k})$$

Step 7: Covariance update
(33)$$P_{k+1}=(I-K_{k+1}H_{k+1})P_{k+1|k}$$

In Equations (20)–(26), $\Phi_{k}$ and $H_{k}$ denote the Jacobian matrices derived from the decrease in $f\left[X_{k}\right]$ and $h\left[X_{k}\right]$ in the $X_{k}$ direction, respectively, $P_{k+1}$ denotes the state variance after the state is updated, and $I_{3}$ stands for the three-order identity matrix.

The above seven steps constitute a calculation cycle of the EKF [23,24,25,26]. When the initial value and the output data of sensors are known, the motion data can be obtained by Equations (20)–(26).

## 4. Simulation of Attitude Measurement Algorithm

### 4.1. Simulation of Attitude Measurement Algorithm Based on Geomagnetic Field

#### 4.1.1. Simulation Parameters

In order to verify the feasibility and accuracy of the solving method based on the geomagnetic field, numerical simulation was performed. According to the geomagnetic model of the terminal-sensitive projectile motion presented in literature [10], the equation for generating the output data of the three-axis magnetic sensor was derived, as given in Equation (14).

In the actual data measurement, there is certain noise in data measured by a sensor. In order to simulate the output data of a three-axis magnetic sensor more realistically, the scanning angle and the scanning angular velocity added with the Gaussian white noise were used as the output data of the three-axis magnetic sensor. The output data of the three-axis magnetic sensor generated by MATLAB simulation are shown in Figure 5.

#### 4.1.2. Simulation Results

The simulation data with the Gaussian white noise were used as the input data of the attitude solving algorithm based on the geomagnetic field. The scanning angle and the scanning angular velocity obtained in the simulation are shown in Figure 6 and Figure 7, respectively. As shown in Figure 6 and Figure 7, the scanning angle and the scanning angular velocity obtained by the algorithm coincided well with the designed scanning parameters; the scanning angle was 30°, and the scanning angular velocity was 4 rev/s. The simulation results showed that the attitude solving algorithm based on the geomagnetic field could accurately calculate the scanning motion parameters of the terminal-sensitive projectile, and the solution results verified the feasibility and accuracy of the proposed algorithm.

In Figure 7, the unit (rev/s) means “revolution/second,” where 1 rev = 360°.

### 4.2. Semi-Physical Simulation

In order to verify the feasibility of the proposed algorithm and the accuracy of the results, the high-precision turntable was used to simulate the scanning motion, and the attitude measuring device was used to collect the accelerometer, gyroscope, and magnetic sensors’ data. The results obtained by the proposed algorithm were compared with the simulation parameters of the turntable.

#### 4.2.1. Turntable Verification Experiment

In the semi-physical simulation experiment, the high-precision three-axis simulation turntable was used to simulate the scanning motion of a terminal-sensitive projectile. The high-precision three-axis turntable used in the semi-physical simulation is shown in Figure 8a. In the simulation experiment, the measurement device was placed on the inner frame and the outside frame rotated; the scanning angle was 32° and the scanning angular velocity was 2 rev/s, as shown in Figure 8b.

#### 4.2.2. Turntable Experiment Results

After reading the data from the attitude measurement device installed on the simulation turntable, the scanning angle and scanning angular velocity were obtained by the proposed algorithm, and the obtained results are presented in Figure 9 and Figure 10, respectively.

According to the semi-physical simulation results, the scanning angle converged to about 32°, the scanning angular velocity converged to about 2 rev/s, and the result coincided well with the simulation parameters. The results of the semi-physical simulation experiment showed that the proposed algorithm had high feasibility and accuracy.

## 5. Airdrop Experiment

In order to simulate the motion environment of the terminal-sensitive projectile more realistically, an unmanned aerial vehicle (UAV) was used in the airdrop experiment. The parachute scanning platform with the installed attitude measuring device was transported to the specified altitude by the UAV for the purpose of airdropping. After the experiment, the data recorded by the attitude measuring device were collected, and the proposed algorithm was used to process the collected data. The result was compared with the designed parameters. At the same time, the scanning angle, scanning angular velocity, and falling velocity of the parachute scanning platform were verified by video analysis.

### 5.1. Airdrop Experiment Procedure

The experiment used the UAV to throw the parachute scanning platform carrying the attitude measuring device from a height of 300 m. In addition, the designed parachute scanning platform had a scanning angular velocity of 4 rev/s and a scanning angle of 30°.

The schematic diagram of the experimental process is shown in Figure 11.

The airdrop test steps were as follows:

Power on the parachute scanning platform, turn on the system, and operate the UAV to carry the parachute scanning platform to a height of 300 m.

The UAV releases the parachute scanning platform once it is stabilized; the parachute scanning platform is put into the free-fall motion and it opens the rotating umbrella.

After the rotating umbrella is fully opened, the umbrella begins to rotate and drives the parachute scanning platform to rotate; during this period, there are accelerating and decelerating processes until the platform reaches a stable rotating scanning state.

The parachute scanning platform enters the steady-state scanning state and stays in that state until it hits the ground and waits for recycling.

### 5.2. Experimental Parameters

The attitude measuring device started to record the data of each sensor when it got power and stopped recording the data when the UAV landed and the device lost power. The parachute scanning platform attitude measurement device used in the experiment is shown in Figure 12.

The measurement device consisted of the main board and four attachment plates, as shown in Figure 13. The main board included two Advanced RISC Machine (ARM) microprocessors, a field programmable gate array (FPGA) microprocessor responsible for logic control, a micro-electromechanical system (MEMS) accelerometer, a yaw rate gyroscope with vibration suppression, a two-axis magnetic sensor, four eight-channel analog-to-digital converters, a flash memory chip, four serial transceiver controllers, related signal processing circuits, and power supply circuits. In addition, the three-axis accelerometer used an American ADI Inc. ADXL377 sensor with a measurement range of ±200 g and shock survivability of 10,000 g. The gyroscope used an American ADI Inc. ADXRS649 sensor with a measurement range of ±20,000°/sec and shock survivability of 10,000 g. The magnetic sensor was an HMC1052L sensor with a measurement range of ±6 gauss and a resolution of 120 microgauss. Four attachment plates were vertically distributed around the main board. Each attachment plate had a two-axis magnetic sensor and a yaw rate gyroscope, where the magnetic sensor was an HMC1052L sensor. The gyroscope used an American ADI Inc. ADXRS646 sensor with a measurement range of ±300°/sec and a shock survivability of 10,000 g. The multi-magnetic sensor could amplify the signal and eliminate the errors. The gyroscopes placed symmetrically could eliminate the errors caused by forward and reverse rotations of the device. The FPGA was responsible for logic control and was equipped with a 16-bit high-precision synchronous analog to digital (AD) converter for achieving high-speed, real-time, and high-quality data acquisition. The 32-bit ARM with Cortex-M7 was responsible for data processing, enabling complex sensor compensation. The data processing module was responsible for human–computer interaction and scheduling relevant calculation results, self-test results, command response, and other operations. The data processing module and the FPGA were connected via the Flexible Memory Controller (FMC) bus to achieve efficient data acquisition. In addition, the device used the advanced Single-Level Cell (SLC) flash particles and HdntRec core independently developed to achieve differential dual-mode data recording.

The signals of the geomagnetic data acquisition module, gyroscope data acquisition module, and acceleration data acquisition module were transmitted to the AD converter after being amplified by the high-speed amplifier. After the AD conversion, the data were transmitted to the data processing module for arithmetic processing. Finally, the processed data were saved to the data storage module. The saved data could be read from the host computer through the communication module, and the motion posture information of the device was obtained after the analysis and inversion.

The results showed that the proposed device could reduce the error of traditional attitude measurement devices such as a gyroscope and a magnetic sensor. At the same time, it could realize high-speed processing and storage of the collected data. The proposed device has the advantages of simple structure, fast startup speed, strong stability, high measurement accuracy, fast processing speed, and the ability to store and read data. Therefore, it is suitable for the flight attitude measurement and analysis of a parachute scanning platform in practice.

### 5.3. Experimental Results

After the airdrop experiment, the data stored in the attitude measurement equipment were read and the output data of each sensor were plotted. The output data of the three-axis accelerometer, three-axis gyroscope, and three-axis magnetic sensor collected by the device in the UAV airdrop experiment are shown in Figure 14, Figure 15 and Figure 16, respectively.

According to the above-presented experimental process and the sensor output data graph, a rough analysis of the data was conducted and the main observations were as follows:

From zero to 2 s, the parachute scanning platform was in a state of free fall and umbrella opening.

From 2 s to 3.5 s, the parachute scanning platform completed the umbrella opening process and started to rotate.

From 3.5 s to 4.5 s, the parachute scanning platform was decelerating and spinning.

At 4.5 s, the steady-state scanning state was basically reached.

From 4.5 s to 18 s, the parachute scanning platform was in a steady-state scanning state with slight fluctuations during the period.

At approximately 18 s, the parachute scanning platform touched the ground.

During the fall of the terminal-sensitive projectile, its movement state was affected by external factors such as wind speed. Thus, the obtained data fluctuated to a certain extent, for instance, for the data at approximately 13 s. The results in Figure 17, Figure 18, Figure 19 and Figure 20 match the movement law of the terminal-sensitive projectile well.

After obtaining the data of each sensor, the scanning angle, scanning angle velocity, falling velocity, and 2D scanning attitude were obtained by the proposed algorithm, and the obtained results are presented in Figure 17, Figure 18, Figure 19 and Figure 20.

According to the experimental results, in the steady-state scanning phase the scanning angle was approximately 30°, the scanning angular velocity was approximately 4 rev/s, the falling velocity was stabilized at approximately 12 m/s, and the 2D scanning attitude conformed to the movement law of the rotation and scanning of the terminal-sensitive projectile. The results of this experiment coincided well with the scanning motion law of the terminal-sensitive projectile and were basically consistent with the set parameters. Consequently, it can be concluded that the proposed method can be used for attitude measurement of parachute scanning platforms and can achieve high measurement accuracy.

### 5.4. Experimental Results Discussion

#### 5.4.1. Variance Analysis

The variance refers to a statistical measurement of the spread between numbers in a data set. More specifically, variance measures how far each number in the set is from the mean and thus from every other number in the set. Variance is often depicted by this symbol: *σ*2. It is used to determine volatility. The square root of the variance is the standard deviation (*σ*), which helps to determine the consistency of error over a period of time. The attitude angle variance curve and the velocity variance curve of the parachute scanning platform are shown in Figure 21.

According to the presented variance curves, the velocity variance was less than 0.03 m/s and the attitude angle variance was less than 0.03°. In addition, the velocity variance curves and the attitude angle variance curves tended to converge. Therefore it can be concluded that with the passage of time, the velocity error and attitude angle error tend to be stable.

#### 5.4.2. Video Data Analysis

During the experiment, a high-definition camera was used to record the motion of the parachute scanning platform. After the experiment, the video data were processed and analyzed to obtain the average scanning angle [27], scanning angular velocity, and falling velocity [28] in the stable phase. The video data analysis results and algorithm results in the period of 14 s to 17 s are given in Table 1.

According to the data presented in Table 1, the following conclusions were drawn. The error between the average scanning angular velocity obtained by the video analysis and the average scanning angular velocity obtained by the algorithm was 0.01rev/s. Similarly, the error of scanning angle was 0.292°, and the error of falling velocity was 0.868m/s. Within the allowable error range, the scanning angle, scanning angular velocity, and falling velocity obtained by the video analysis were equal to the corresponding parameters obtained by the algorithm. The proposed attitude measurement method based on the combination of a magnetic sensor and an inertial sensor could measure the attitude of the terminal-sensitive projectile accurately, and the measurement results had a high reference value.

In terms of terminal-sensitive projectile attitude measurement, the traditional measurement method has the following shortcoming: The calculation accuracy is not high, a group of experiments can obtain only a single or a small number of scanning parameters, and measurement results are easily affected by external interference. In contrast, the proposed attitude measurement scheme can accurately measure multiple scanning motion parameters from a group of experiments, including the scanning angle, scanning angular velocity, falling velocity, and 2D scanning attitude. Simultaneously, the optimal combination and layout of multiple sensors can improve the reliability, anti-interference ability, and measurement accuracy of the device.

## 6. Conclusions

This paper proposes a combined measurement method based on inertial and geomagnetic sensors, which can accurately and simultaneously measure multiple motion parameters of a parachute scanning platform, including the scanning angle, scanning angular velocity, and falling speed, thus significantly improving the measurement efficiency. Moreover, the proposed method adopts a combined measurement method, and the system stability and anti-interference ability are improved. The proposed method was verified by experiments, and according to the experimental results, the proposed combined measurement method can overcome the shortcoming that the traditional measurement method of a parachute scanning platform can obtain only one signal measurement parameter at a time. In addition, the measurement results obtained by the proposed method have high accuracy, which can meet the measurement requirements of the parachute scanning platform. The research results presented in this paper can provide a useful reference for the design of parachute scanning platforms, thus decreasing the development costs and shortening the development time.

The range of the projectile in this study was within 300 m, so some improvements should be developed in higher-range cases. In the future work, it is possible to consider introducing more sensor information and adopting a reasonable filter to obtain a higher-precision projectile attitude.

## Figures and Tables

**Figure 1 micromachines-12-00402-f001:**
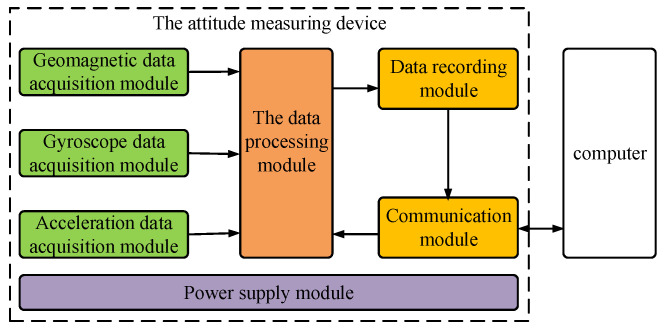
Block diagram of the proposed attitude measuring device.

**Figure 2 micromachines-12-00402-f002:**
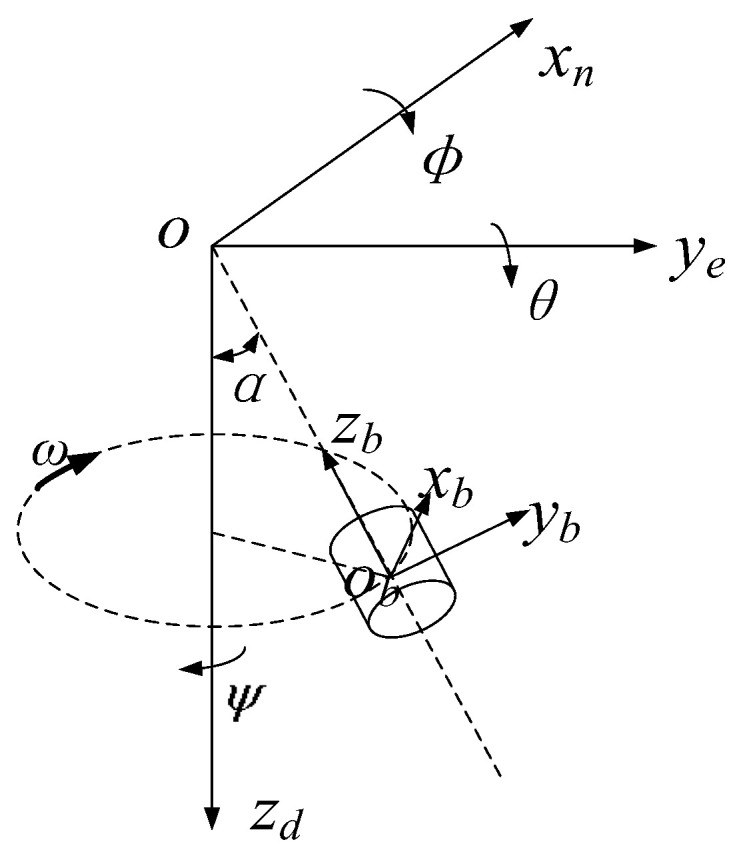
Illustration of navigation coordinate system and projectile coordinate system.

**Figure 3 micromachines-12-00402-f003:**
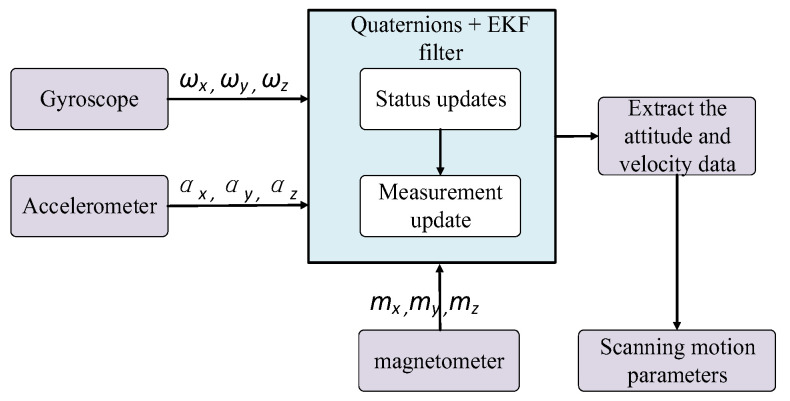
Block diagram of the scanning motion parameter estimation algorithm.

**Figure 4 micromachines-12-00402-f004:**
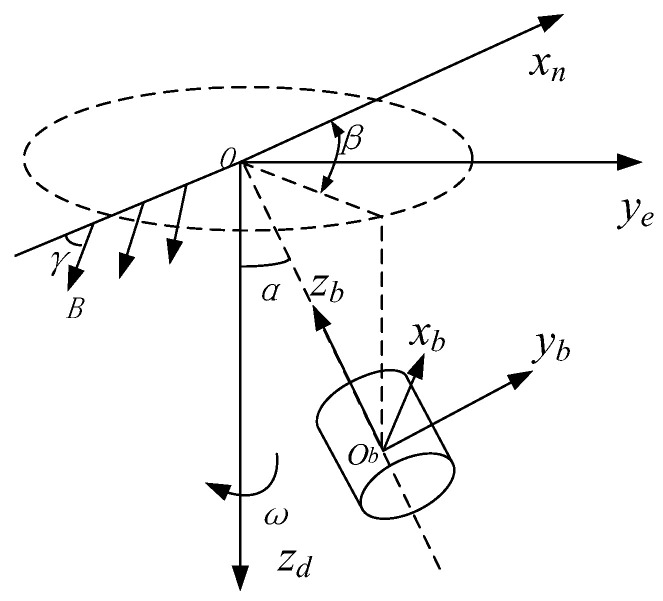
Attitude measurement based on the geomagnetic field.

**Figure 5 micromachines-12-00402-f005:**
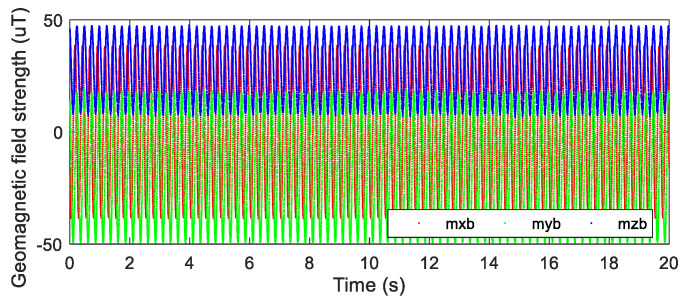
Simulation data of the three−axis magnetic sensor.

**Figure 6 micromachines-12-00402-f006:**
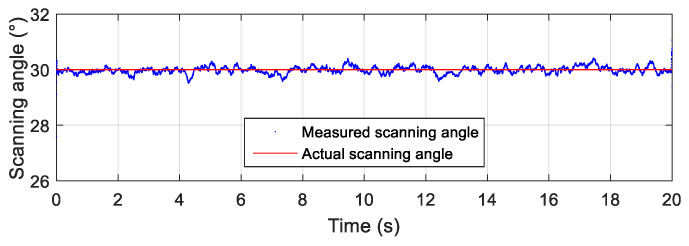
Scanning angle of the simulation based on the geomagnetic field.

**Figure 7 micromachines-12-00402-f007:**
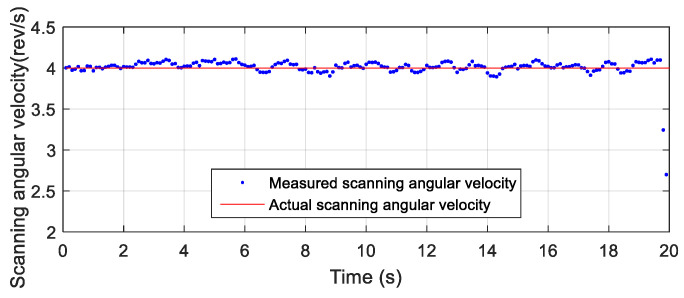
Scanning angular velocity of the simulation based on the geomagnetic field.

**Figure 8 micromachines-12-00402-f008:**
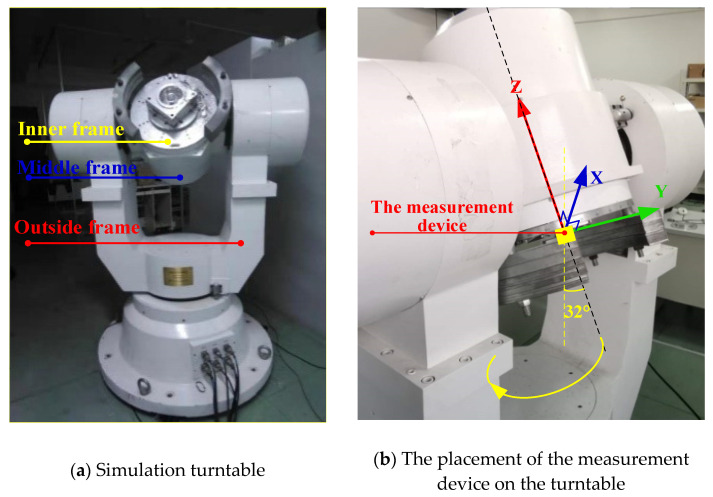
Simulation turntable and the placement of the measurement device.

**Figure 9 micromachines-12-00402-f009:**
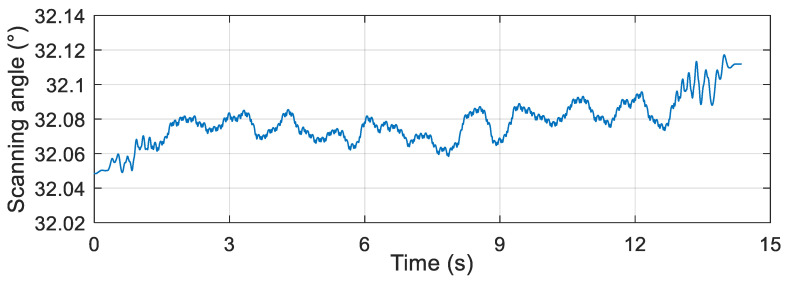
Scanning angle of the semi-physical simulation.

**Figure 10 micromachines-12-00402-f010:**
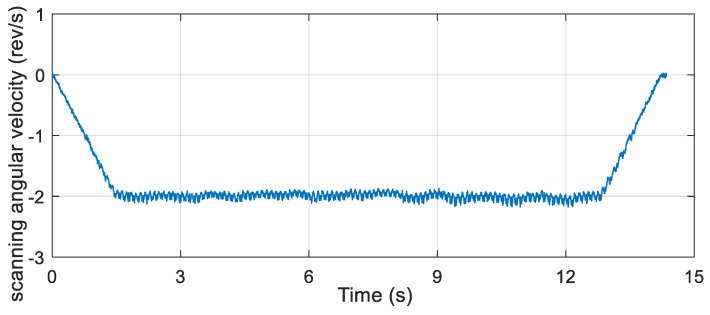
Scanning angular velocity of the semi-physical simulation.

**Figure 11 micromachines-12-00402-f011:**
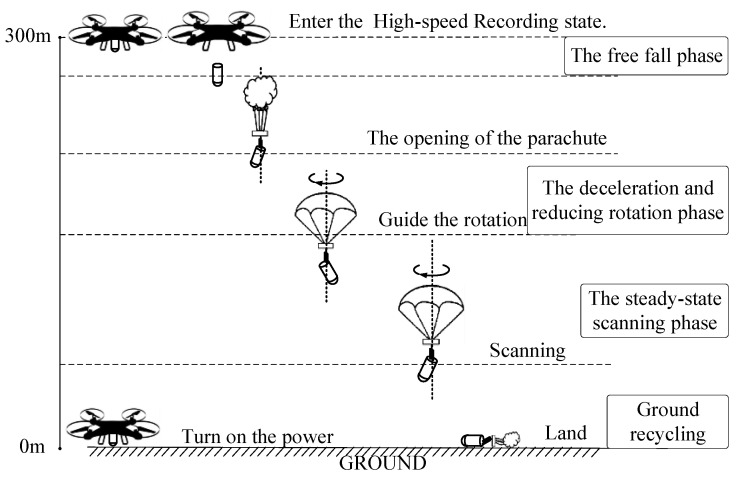
Flow diagram of the airdrop experiment.

**Figure 12 micromachines-12-00402-f012:**
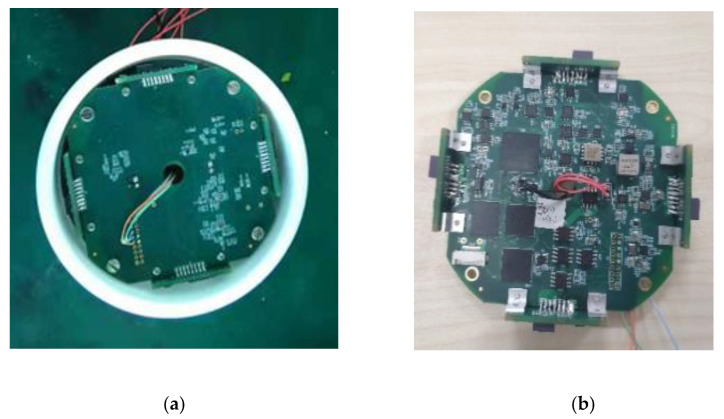
The parachute scanning platform attitude measurement device. (**a**) The bottom of the circuit; (**b**) The top of the circuit.

**Figure 13 micromachines-12-00402-f013:**
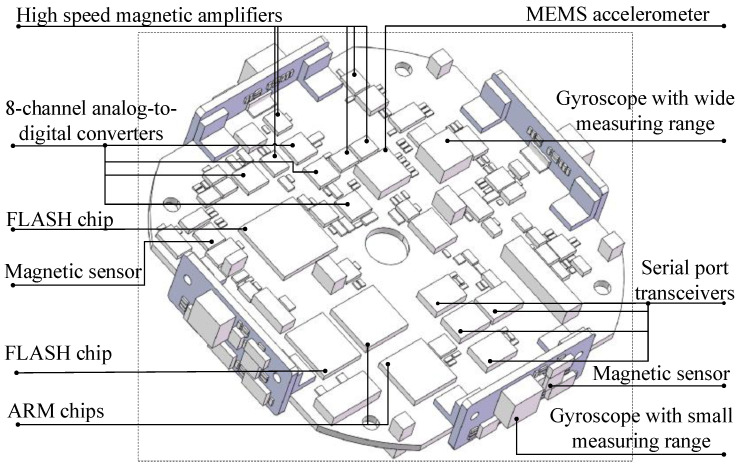
The structure of the parachute scanning platform attitude measurement device.

**Figure 14 micromachines-12-00402-f014:**
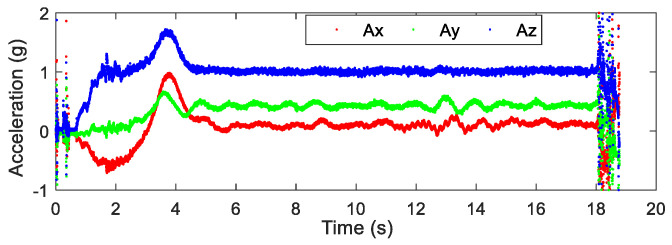
Output data of the three-axis accelerometer.

**Figure 15 micromachines-12-00402-f015:**
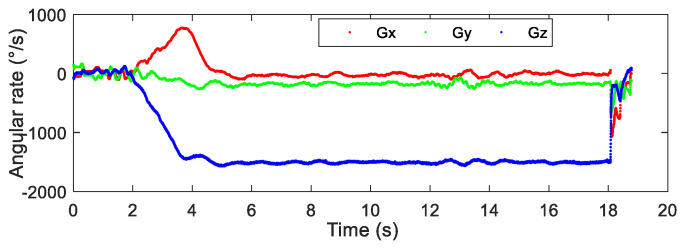
Output data of the three-axis gyroscope.

**Figure 16 micromachines-12-00402-f016:**
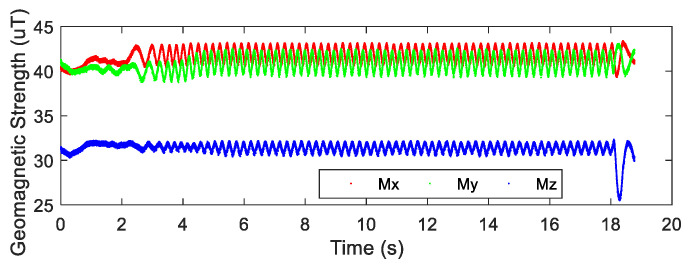
Output data of the three-axis magnetic sensor.

**Figure 17 micromachines-12-00402-f017:**
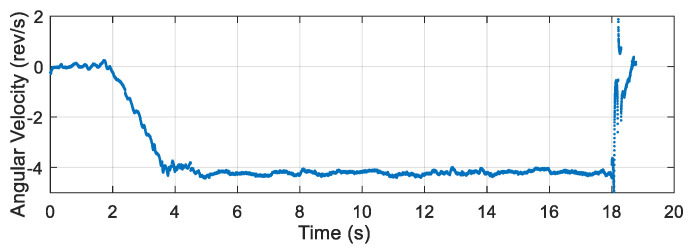
Scanning angular velocity of the airdrop experiment.

**Figure 18 micromachines-12-00402-f018:**
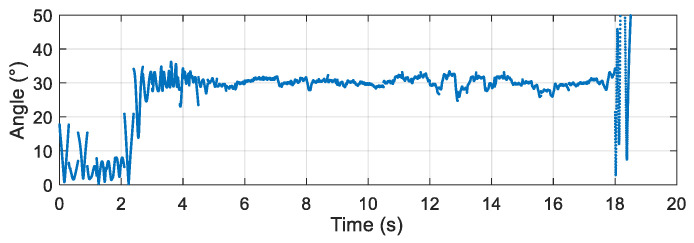
Scanning angle of the airdrop experiment.

**Figure 19 micromachines-12-00402-f019:**
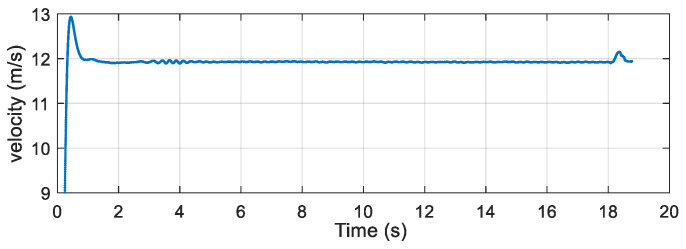
Falling velocity of the airdrop experiment.

**Figure 20 micromachines-12-00402-f020:**
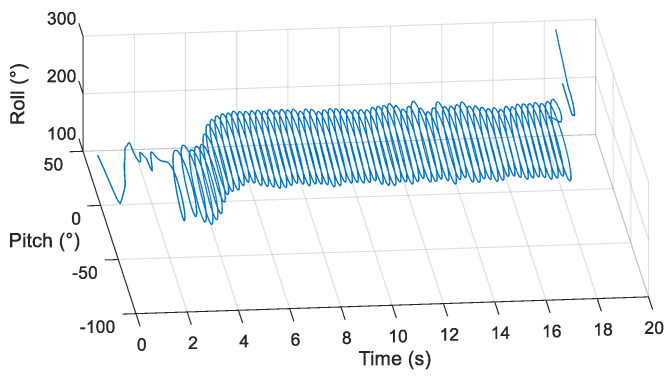
2D scanning attitude of the airdrop experiment.

**Figure 21 micromachines-12-00402-f021:**
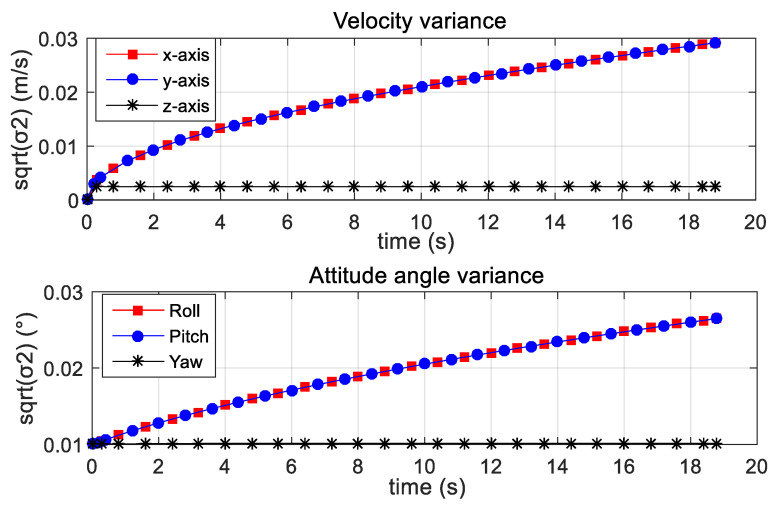
Velocity variance and attitude angle variance results.

**Table 1 micromachines-12-00402-t001:** The comparison of video data and solution data.

Variable	Source	Value
Average scanning angular velocity (rev/s)	video	4.194
	algorithm	4.204
Average scanning angle (°)	video	29.744
	algorithm	30.036
Average falling velocity (m/s)	video	11.034
	algorithm	11.902

rev = revolution.

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
