# Peer review of "Research on Measurement Method of Parachute Scanning Platform Based on MEMS Device"

_micromachines, 2021, doi:10.3390/mi12040402_

Round 1

Reviewer 1 Report

This paper by Liu et al reports a sensing system for measuring the attitude of an object descending with a parachute. This system uses an inertial measurement unit (IMU) which combines a magnetic sensor, gyroscope, and accelerometer. After the data acquisition, an EKF filter is applied to reduce the noise. The system is verified by an experiment with a three-axis turntable and by an air-drop experiment. The method itself of using those sensors and the EKF filter is already a popular sensing method for unmanned systems [10.1109/TNSRE.2005.847353, 10.1109/MFI.2010.5604460], and they applied this system to a parachute scanning platform aiming for terminal-sensitive projectiles.

I see six issues that the authors should address before publication.

  • Plots in Figures 9, 10, and 14-20 are plotted with lines although they are experimental data. I suggest using dots instead to express the time interval of the measurement.
  • The y-axis in Figure 16 is “Geomagnetic strength” and has no unit. The parameter can be converted to the magnetic field (maybe μT).
  • The time scale in figures 9, 10, and 14-20 is difficult to understand because they are using a millisecond unit with “×104”. Instead, using a second unit can help readers to understand.
  • There is a typo in Figure 3, “velocity”.
  • The authors successfully applied IMU to the parachute scanning platform, but the originality of this application is not well presented. If this paper is applying an existing method to a system similar to the one used before without an original idea, the originality will be low.
  • Many of the cited papers [1-10] about terminal-sensitive projectile were dissertations or Chinese local papers that were not accessible for the reviewer.

Author Response

Dear Reviewer,

Thank you for allowing a resubmission of our manuscript, with an opportunity to address the reviewers’ comments. We have addressed the concerns and criticisms of the reviewers in detail. Simultaneously, we have tried to check and revise the grammar of the full text more carefully in the revision.

We are uploading (a) our point-by-point response to the comments (below) (response to reviewers), (b) an updated manuscript with yellow highlighting indicating changes, and (c) a clean updated manuscript without highlights.

Best regards,

Ning Liu et al.

Reviewer 2 Report

Please explain the unit (B) in Fig.5

In fig.5 the time range is 2 s whereas in fig.6 and 7 - 20 s , why?

Also explain the unit in fig.7

In fig.16 unit in vertical axis?

The fig.21 and table 1 should be better explained

Author Response

(The authors gave the same response as above.)

Round 2

Reviewer 1 Report

In the revised manuscript, Liu et al. fixed the expressions and figures pointed by the reviewer and also cited journal papers that are accessible for the reviewer. However, their originality and significance are not yet clearly revealed in their introduction because they are not citing other works to compare to this study.

If "the method of calculating the scanning parameters of parachute scanning platform based on geomagnetism " is their original idea, as the author said, they may be able to present other studies that modeled and calculated the descending motion of the object with a parachute. For example, many studies assume pendulum motion rather than the scanning motion [1-2]. The author may be able to present the strong point or suitable situation of assuming scanning motion.

[1] DOI: 10.2514/1.4783 
[2] DOI: 10.2514/6.2015-2138

Author Response

Original Manuscript ID: micromachines-1133566     

Original Article Title: “Research on Measurement Method of Parachute Scanning Platform Based on

MEMS Device

To: Micromachines Editor

Re: Response to reviewers

Dear Editor,

Thank you for allowing a resubmission of our manuscript, with an opportunity to address the reviewers’ comments. We have addressed the concerns and criticisms of the reviewers in detail. Simultaneously, we have tried to check and revise the grammar of the full text more carefully in the revision.

We are uploading (a) our point-by-point response to the comments (below) (response to reviewers), (b) an updated manuscript with yellow highlighting indicating changes, and (c) a clean updated manuscript without highlights.

Best regards,

Ning Liu et al.

This manuscript is a resubmission of an earlier submission. The following is a list of the peer review reports and author responses from that submission.